# Axonal Modulation of Striatal Dopamine Release by Local γ-Aminobutyric Acid (GABA) Signalling

**DOI:** 10.3390/cells10030709

**Published:** 2021-03-23

**Authors:** Bradley M. Roberts, Emanuel F. Lopes, Stephanie J. Cragg

**Affiliations:** Department of Physiology, Anatomy and Genetics, Centre for Integrative Neuroscience and Oxford Parkinson’s Disease Centre, University of Oxford, Oxford OX1 3PT, UK

**Keywords:** dopamine, GABA, GABA_A_ receptors, GABA_B_ receptors, GABA transporters, Parkinson’s disease, striatum, tonic inhibition

## Abstract

Striatal dopamine (DA) release is critical for motivated actions and reinforcement learning, and is locally influenced at the level of DA axons by other striatal neurotransmitters. Here, we review a wealth of historical and more recently refined evidence indicating that DA output is inhibited by striatal γ-aminobutyric acid (GABA) acting via GABA_A_ and GABA_B_ receptors. We review evidence supporting the localisation of GABA_A_ and GABA_B_ receptors to DA axons, as well as the identity of the striatal sources of GABA that likely contribute to GABAergic modulation of DA release. We discuss emerging data outlining the mechanisms through which GABA_A_ and GABA_B_ receptors inhibit the amplitude as well as modulate the short-term plasticity of DA release. Furthermore, we highlight recent data showing that DA release is governed by plasma membrane GABA uptake transporters on striatal astrocytes, which determine ambient striatal GABA tone and, by extension, the tonic inhibition of DA release. Finally, we discuss how the regulation of striatal GABA-DA interactions represents an axis for dysfunction in psychomotor disorders associated with dysregulated DA signalling, including Parkinson’s disease, and could be a novel therapeutic target for drugs to modify striatal DA output.

## 1. Introduction

Release of the neurotransmitter dopamine (DA) in the brain is critical for action selection, motivation and cognition and is dysregulated across a diversity of disorders including Parkinson’s disease (PD) and addictions. Forebrain DA originates primarily from midbrain DA neurons in the substantia nigra pars compacta (SNc) and the ventral tegmental area (VTA). Axons from these neurons course through the medial forebrain bundle (MFB) to provide rich DA innervation of the striatal complex, comprising the dorsal striatum (caudate putamen, CPu) and the nucleus accumbens (NAc) core and shell, and more limited innervation of other basal ganglia nuclei, including the subthalamic nucleus and globus pallidus. Within the striatum, mesostriatal DA axons form immense unmyelinated axonal arbours: the axonal bush formed by one nigrostriatal neuron forms an average 470 µm total axon length, with ~10^4^ branch points and ~10^5^
*en passant* DA release varicosities, and covers a mean of 2.7% of the total volume of the striatum in rats [1,2,3,4]. These axonal attributes are probably unique in the CNS to DA neurons, rivalled only in length (but not branching) e.g., by basal forebrain cholinergic neurons [5]. DA axon arbours could be major strategic sites for striatal inputs to influence axonal propagation of action potential and DA output through mechanisms distinct from those governing action potential generation at the level DA soma in midbrain [6,7,8]. DA release in the striatum can be gated locally by a variety of striatal neuromodulators [6,7,8,9], which can even independently drive DA release, altogether demonstrating that direct modulation of the DA axon is a powerful means of determining striatal DA output in a manner that is independent of somatic processing [10,11,12,13].

The striatum contains a high density of neurons that release the inhibitory neurotransmitter γ-aminobutyric acid (GABA), that comprise principally spiny projection neurons (SPNs) (~95%), and a diversity of GABAergic interneurons (~2–3%) including fast-spiking interneurons (FSIs), low-threshold spiking interneurons (LTSIs), calretinin-expressing interneurons, tyrosine hydroxylase-expressing interneurons, neurogliaform interneurons, fast-adapting interneurons and spontaneously active bursty interneurons [14,15]. In addition, GABA may be co-released from DA axons and cholinergic interneurons (ChIs) [16,17] and a small population of GABAergic neurons in the SNc and VTA project to the striatum [18,19]. The volume of striatum reached by an average rat nigrostriatal DA neuron arbour (2.7%) [1] contains ~74,000 GABAergic neurons, calculated from 2.8 million striatal neurons per hemisphere, of which ~98% are glutamic acid decarboxylase (GAD)-immunoreactive [20]. While it is well understood that the release of DA from mesostriatal DA axons in striatum will modulate the activity of many of these striatal GABAergic neurons, both directly through DA receptor signalling and indirectly through facilitating corticostriatal plasticity [21,22], what is less well known is that the reciprocal relationship also occurs, whereby local GABA signalling also modulates striatal DA release. To date, no GABAergic axoaxonic synapses have been identified on DA axons [23], but a wealth of historical and more recently refined evidence has revealed that local GABA signalling in striatum can powerfully modulate DA release through action at GABA_A_ and GABA_B_ receptors. We review these actions, sources and substrates, in detail here.

## 2. GABA_A_ and GABA_B_ Receptor-Mediated Inhibition of Striatal DA Release

There is an assortment of historical but conflicting evidence indicating that striatal GABA can locally and bi-directionally modulate DA output. One of the earliest studies perfused GABA (10^−5^ M) into the caudate nucleus of anaesthetised cats, and found an initial potentiation followed by prolonged inhibition of the release of radiolabelled ^3^H-DA synthesised from l-3,5-^3^*H*-tyrosine measured from the superfusate sampled from push-pull cannulae [24]. Contemporaneous studies in rat striatal slices similarly demonstrated bidirectional effects of exogenous GABA on the release of radiolabelled ^3^H-DA in striatum, with low concentrations of GABA (10^−5^–10^−3^ M) potentiating, and higher concentrations of GABA (10^−1^ M) inhibiting, the spontaneous and potassium-stimulated release of DA [25,26]. Similar bi-directional effects were seen in DA release evoked by electrical stimulation in striatal slices prepared from rabbit [27]. The potentiation of striatal DA levels induced by infusions of low concentrations of GABA in these early studies, however, was only mildly occluded [25,26], or not at all modified [24,27], by the presence of GABA_A_ receptor antagonists bicuculline or picrotoxin. When GABA uptake transporters were blocked, the effects of low concentrations of exogenous GABA (10^−4^–10^−3^ M) were reversed, resulting in a prevailing inhibition of striatal radiolabelled ^3^H-DA release evoked by electrical stimulation [27]. These early studies indicated that striatal DA release could be modulated by local GABA signalling, but the mechanisms and direction of impact to either potentiate or inhibit striatal DA release remained unresolved.

Subsequent studies arising from the early 1990s onwards, including those employing more refined electrochemical techniques to detect endogenous DA, and on rapid time-scales, revealed that an endogenous striatal GABA source could locally modulate DA output through action at striatal GABA_A_ and GABA_B_ receptors. Striatal administration of GABA_A_ or GABA_B_ receptor agonists or positive allosteric modulators decreased extracellular DA levels in striatum of intact rats measured by microdialysis in vivo, while conversely, GABA_A_ and GABA_B_ antagonists independently increased extracellular DA levels [28,29]. GABA_A_ antagonists were also shown to increase NMDA-evoked ^3^H-DA release synthesised by l-3,5-^3^*H*-tyrosine in ex vivo striatal rat brain slices [30] and GABA_A_ agonists were shown to inhibit potassium-stimulated ^3^H-DA release from rat striatal synaptosomes [31]. Studies employing fast-scan cyclic voltammetry to detect DA revealed an apparently conflicting GABA_A_-mediated potentiation of DA release in acute ex vivo striatal guinea pig slices during prolonged electrical pulse train stimulations, which was shown to be mediated indirectly via inhibition of H_2_O_2_ release from SPNs during train stimulations and a consequent disinhibition of DA release [32,33]. Subsequent studies with fast-scan cyclic voltammetry and using shorter, single electrical or optogenetic stimulation of ChR2-expressing DA axons in acute slices of mouse striatum have shown that striatal administration of GABA_A_ and GABA_B_ agonists attenuate DA across dorsal-ventral striatal territories [34,35,36,37]. These findings in slice preparations demonstrate a local mechanism of GABAergic inhibition of DA release in striatum, and are less confounded than in vivo studies by potential effects on extrastriatal circuits e.g., effects on striatal DA release via changes in midbrain DA neuron firing. Moreover, very recent studies in striatal slices have revealed that endogenous striatal GABA, across dorsal-ventral striatal territories, provides a tonic inhibition of DA release. When short single optogenetic stimulus pulses were used to activate ChR2-expressing DA axons and limit co-activation of other local neurons [38], both GABA_A_ and GABA_B_ receptor antagonists independently enhanced DA release [36,37,39]. Altogether, these findings demonstrate that endogenous GABA locally released in striatum can act at both GABA_A_ and GABA_B_ receptors to inhibit DA release, and this inhibition can operate tonically.

GABA receptor activation decreases the amplitude of extracellular DA concentrations released by a discrete stimulus when detected from a population of release sites by an extracellular probe, presumably by decreasing the release probability of dopamine per site and consequently the total number of sites that release DA in response to the stimulus. GABA receptor activation also has modest but significant effects on the frequency sensitivity of DA release. GABA_A_ and GABA_B_ receptor activation interacted with frequency and number of stimulus pulses to slightly promote the ratio of DA release evoked by high-frequency trains over single pulse release (see Figure 1) [36,37,39]. This outcome is consistent with a reduction in the initial release probability of DA, and consequentially, a slight relief of short-term depression. Therefore, GABA receptor activation not only primarily limits the overall amplitude of DA output but additionally mediates a minor enhancement in the frequency filtering. GABA tone could therefore promote the relative change in DA output in response to ascending action potential from DA neuron activity, and so enhance the contrast in DA signals seen between high-frequency bursts of action potential versus low-frequency tonic firing. These effects of GABA on frequency filtering of DA release are modest, however, when contrasting with the effects of striatal ACh released from ChIs, where ACh is a major player in more strongly gating the frequency sensitivity of evoked DA release [40,41].

## 3. Direct vs. Indirect Actions of GABA_A_ and GABA_B_ Receptors That Inhibit DA Release

It is incompletely resolved whether the striatal GABA receptors that inhibit DA release are located directly on DA axons, or act indirectly by impacting on other striatal circuits that modulate DA release. However, current evidence, which we review in this section, strongly suggests action of GABA at receptors on DA axons.

At the level of the midbrain, DA neurons in the substantia nigra and VTA can be immunolabelled for GABA_A_ and GABA_B_ receptors [42,43,44], which promote DA neuron hyperpolarisation and/or inhibition of firing [45,46]. At the level of DA axons in striatum, an ultrastructural immunohistochemical study in the monkey striatum demonstrated expression of GABA_B_ receptors on structures that resemble DA axons [23]. However, conclusive anatomical evidence for GABA_A_ and GABA_B_ receptors located on confirmed DA axons has not yet been reported. Direct electrophysiological recordings of nigrostriatal axons strongly support the presence of functional GABA_A_ in DA axons [47], as do the findings that GABA_A_ and GABA_B_ agonists and antagonists can respectively suppress and enhance DA release when evoked by discrete single electrical or optogenetic stimulus pulses of ChR2-expressing DA axons [34,36,37,39]. These brief (≤2 ms) and targeted stimulation paradigms would offer very limited opportunity for activation of other striatal circuits to impact on DA released by the same stimulus. Furthermore, a candidate mechanism whereby GABA receptors on glutamate afferents operate as an intermediary is unlikely, as the enhancement of evoked DA release by GABA receptor antagonists does not require striatal glutamate receptors [39]. A mechanism involving GABA receptors located on GABAergic interneurons is also unlikely, as GABA receptor agonists would be expected to suppress GABAergic interneurons and therefore enhance DA output.

We previously explored whether ChIs might be a potential locus to indirectly mediate the GABAergic inhibition of DA release [36]. Striatal ChIs play a particularly powerful role in gating DA release through nicotinic acetylcholine receptors (nAChRs) on DA axons [40,41,48], and can mediate the effects of other striatal neuromodulators on DA release, including opioids, nitric oxide, glutamate, corticotropin releasing factor and insulin [49,50,51,52,53]. Striatal ChIs are known to express both GABA_A_ and GABA_B_ receptors [54,55] and receive local and extrastriatal GABA inputs which are capable of modulating ChI excitability [56], making ChIs a compelling locus to indirectly mediate the GABAergic inhibition of DA release. However, inhibition of nAChRs did not prevent the effect of GABA receptor agonists and antagonists on electrically evoked DA release, indicating that GABA action on ChIs, and nAChR activation, is not required for GABAergic modulation of DA release [34,36]. Furthermore, the effects of GABA receptor agonists and antagonists on DA release evoked by optogenetic stimulation of ChR2-expressing DA axons bypass ChIs as this mode of DA release is independent of nAChR input [13,38]. These findings rule out indirect GABAergic modulation of DA release via ChI action at nAChRs on DA axons, but ChIs might nonetheless contribute a GABA source to inhibit DA release, as approximately half of all ChIs are GAD-positive and thought to be capable of co-releasing GABA [17], see Section 6.

Together, the functional evidence to date suggests that the primary mechanism of GABA-mediated inhibition of DA release is very likely through direct action at GABA_A_ and GABA_B_ receptors located on DA axons themselves. However, the anatomical location of these receptors on DA axons remains to be confirmed ultrastructurally. Furthermore, until all of the striatal sources of neuromodulatory control of DA are resolved, it remains possible that GABA-receptors on other neurons and inputs might mediate the inhibitory control of DA output.

## 4. GABA_A_ Receptor Modulation of DA Axonal Processing

GABA_A_ receptors are pentameric ligand-gated ion channels that undergo a conformational change upon binding two agonist molecules, subsequently increasing membrane permeability to chloride ions. Increasing evidence indicates that GABA_A_ receptors are prevalent not only in the somatodendritic compartments of CNS neurons, but also in their axonal compartments. GABA_A_ receptors are generally hyperpolarising at postsynaptic sites on somatodendrites, where the GABA_A_ reversal potential is negative relative to the resting membrane potential, and therefore postsynaptic GABA_A_ receptors principally exert an inhibitory action on downstream neurotransmission. This concept is not comprehensive however, as somatodendritic GABA_A_ receptors can be depolarising during adolescence in many neuron types e.g., [57], and can also be depolarising in specific somatodendritic compartments and in particular neuron types in adulthood, e.g., the dendrites or axon initial segment of pyramidal neurons [58]. At the level of the axon, local GABA_A_ receptor activation can be either depolarising or hyperpolarising due to the varying axonal resting membrane potentials observed across neuronal cell types and across development [57,59,60,61]. Axonal GABA_A_ receptors have been shown to exert excitatory action on neurotransmission in many CNS neurons across the mammalian brain (for review see [62]). For example, axonal GABA_A_ receptors enhance synaptic transmission from cerebellar granule cell parallel fibres [63,64,65,66], terminals of cerebellar Purkinje neurons [61], hippocampal mossy fibres [60] and in layer 2/3 pyramidal neurons in cortex [67], where GABA_A_ receptor activation is thought to be depolarising on a background of hyperpolarised membrane potentials. By contrast, axonal GABA_A_ receptors have inhibitory action on neurotransmission in ventral horn group Ia and 1b afferents in the spinal cord [68,69,70]. The exact mechanisms by which axonal GABA_A_ receptors can inhibit neurotransmission are incompletely established because of lack of experimental access to the axonal compartment, but proposed mechanisms involve a shunting inhibition resulting from the increase in membrane conductance upon activation of GABA_A_ receptors, or sodium channel inactivation preventing membrane reactivation and thus preventing release [62].

A recent study using whole-cell and perforated-patch recordings to test for GABA_A_ receptors on the axonal segment of DA neurons within the striatum of adult mice found that the application of GABA mediated a GABA_A_ receptor-dependent depolarisation of DA axons [47]. Despite this depolarisation, activation of axonal GABA_A_ receptors results in the inhibition of DA release [36,37,39,47], in contrast to the effects of axonal GABA_A_ receptors elsewhere that enhance synaptic transmission, outlined above. The depolarising effects of GABA_A_ receptors were found to paradoxically inhibit DA release through underlying mediators involving shunting inhibition and depolarisation-mediated inactivation of sodium channels [47]. The observed shunting inhibition and depolarisation-mediated inactivation of sodium channels was shown to inhibit DA release by limiting the amplitude and propagation of axonal action potentials: GABA_A_ receptor agonists had only subtle effects on intracellular calcium levels reported by GCaMP signals in striatal DA axons at sites proximal to the site of stimulation, but profoundly reduced calcium levels in distal axons [47]. Therefore, the voltage-gated sodium channels and other channels that support active propagation of action potentials in DA axons are challenged by axonal GABA_A_ receptor activation, which may have a strong effect in highly branched, unmyelinated and varicose DA axons, to limit DA output.

Axonal GABA_A_ receptors in CNS neurons can be tonically activated by low concentrations of ambient GABA (near µM levels) to generate continuous chloride currents, which exhibit low amplitudes, long decay times and little or no desensitisation [62,71]. Tonic activation of GABA_A_ receptors is largely contingent on the GABA_A_ receptor subunits expressed, with receptors containing the δ subunits mediating tonic GABAergic currents. When δ subunit-containing receptors lack a γ subunit, this promotes plasmamembrane expression of receptors at extrasynaptic locations (for review see [62,71,72]). In situ hybridisation studies have identified mRNA transcripts for α1, α2, α3, α4, β1, β2, β3 and γ2 GABA_A_ receptor subunits in midbrain DA neurons at the level of the soma in human post-mortem tissue and in TH^+ve^ neurons in mice [73,74]. However, the particular GABA_A_ receptor subunits that become targeted for function in striatal DA axons have not yet been identified and could differ from those targeted to the soma level. Interestingly, a recent study has identified a tonic GABAergic inhibitory current at the soma level in VTA DA neurons in mice mediated by an extracellular GABA tone acting at GABA_A_ receptors with a relativity unusual combination of α4βε subunits [75]. Whether these operate in DA axons is not yet known. Information about GABA receptor type would be an important advance for the field, as it would provide an anatomical substrate for the GABA_A_ receptor-dependent modulation of DA release, give insight into tonic activation states and synaptic versus extrasynaptic actions, and potentially, identify a unique pharmaceutical target. Notwithstanding this paucity of information, these receptors are thought to be tonically activated by endogenous GABA. Striatal DA release is under a tonic inhibition through a GABA_A_ (and GABA_B_) receptor-mediated component [29,36,37], and furthermore, diazepam, a positive allosteric modulator of GABA_A_ receptors, decreases the input resistance of striatal axons, indicating an action in concert with a striatal GABA tone [47]. Under conditions of tonic GABA_A_ receptor activation then, the shunting inhibition and depolarisation-mediated inactivation of sodium channels will be prevalent, and particularly pertinent to an electrically tight compartment like the axon where small fluctuations of GABA_A_ activity might cause large changes in membrane potential and input resistance to facilitate strong regulation of information processing across the extensive DA axon arbour.

## 5. GABA_B_ Receptor Modulation of DA Axonal Processing

GABA_B_ receptors are chloride-independent metabotropic receptors that mediate inhibition by heterotrimeric G-protein activation and typically suppress neurotransmitter release through various second-messenger-mediated mechanisms, which include (i) inhibition of calcium influx through voltage-gated calcium channels (VGCCs), (ii) inhibition of adenylyl cyclase resulting in retarded synaptic vesicle recruitment, and (iii) activation of Kir3-type potassium channels leading to membrane hyperpolarisation and shunting of excitatory currents (for extensive review see [76]). Importantly, presynaptic or axonal GABA_B_ receptors, like GABA_A_ receptors, can also be under tonic activation by low concentration extrasynaptic or ambient GABA to tonically gate the probability of neurotransmitter release from many cell types including layer I Cajal-Retzius neurons in cortex [77,78], thalamocortical projection neurons [79], CA3-CA1 hippocampal neurons [80] and cerebellar granule neurons [81]. The precise intracellular mechanisms through which axonal GABA_B_ receptors inhibit the amplitude of DA release is yet to be fully elucidated and could involve any of these standard players.

While GABA_B_ receptors on striatal DA axons function to inhibit the overall amplitude of DA release, which could be consistent with modified VGCC activity, they might also simultaneously gate short-term plasticity of DA release by modifying axonal excitability via potassium-dependent conductances through activation of Kir3-type potassium channels. GABA_B_ receptor activation slightly promotes the ratio of evoked DA release by high-frequency trains over single pulse release (see Figure 1) [36,37,39], consistent with a reduction in the initial release probability of DA, and consequently, a slight relief of short-term depression. We have recently revealed that mechanisms that determine axonal excitability, particularly potassium-dependent processes, strongly gate short-term plasticity of DA release [82]. Tonic activation of GABA_B_ receptors on DA axons might plausibly provide a persistent axonal permeability to potassium through activation of Kir3-type potassium channels, and we therefore hypothesise that tonic GABA_B_ activity might facilitate changes to short-term plasticity of DA release in response to conditions that change the driving force on potassium entry. Future studies will need to determine the precise intracellular mechanisms through which GABA_B_ receptors on DA axons modify striatal DA release, and whether these mechanisms impact on other drivers of short-term plasticity of DA release.

## 6. Sources of Striatal GABA Mediating GABAergic Inhibition of DA Release

The striatal sources of GABA that provide tonic GABAergic inhibition of DA output have not yet been fully elucidated. Here, we discuss the potential sources, which might include a net striatal ambient GABA tone, GABA co-release from mesostriatal DA neurons themselves, GABAergic mesostriatal inputs, tonically active LTSIs, ChIs, and release of GABA from striatal astrocytes.

Synthesis of GABA throughout the mammalian brain, including in striatum, canonically requires GAD. We recently established that the source(s) of GABA responsible for the tonic GABAergic inhibition of DA release originated from a canonical GAD-dependent neuronal source (i.e., striatal GABAergic neurons). Pre-treatment with GAD inhibitor 3-mercaptopropionic acid prevented the disinhibition of striatal DA release by GABA-receptor antagonists [39]. There are GAD-independent GABA sources in striatum, including the GABA co-released from mesostriatal DA neurons that is synthesised by the non-canonical GABA synthesis enzyme aldehyde dehydrogenase (ALDH)-1a1 [83]. ALDH inhibition did not prevent the disinhibition of striatal DA release by GABA-receptor antagonists [39], and therefore the primary source of GABA mediating the tonic inhibition of striatal DA release is GAD-dependent.

To date, no GABAergic axoaxonic synapses have been identified on DA axons [23], although the reciprocal arrangement of TH-immunoreactive DA axons forming direct synaptic contacts with dendrites of GAD-positive GABAergic neurons is documented in rats [84]. Despite the evident close apposition of these two neurotransmitter systems, a candidate synaptic GABAergic input that can serve the GABAergic inhibition of DA release is not yet identified. Furthermore, the lack of identified GABAergic synaptic input to DA axons raises the question of whether GABA tone on DA axons arise from extrasynaptic GABA. GABA can spill over from synapses for extrasynaptic function in many other brain nuclei [71], and in the striatum provides a sizeable ambient GABA tone on spiny projection neurons, evident as a tonic GABA_A_ receptor-mediated inhibitory conductance [85,86,87,88]. This documented ambient GABA tone in striatum is likely not spatially restricted to spiny projection neurons and could feasibly be a potential source mediating the tonic GABAergic inhibition of DA release.

The ambient GABA tone detected in striatum by SPNs is, at least in part, reported to be action potential-independent [39,87,89]. Therefore, striatal ambient GABA tone and tonic inhibition at GABA receptors might arise from extrasynaptic spillover of both action potential-dependent and independent spontaneous GABA release. An ambient GABA tone in striatum arising from a spontaneous GABA source that can act on DA axons to limit DA output should not be entirely surprising when considering the immense GABAergic network contained within the stratal DA axonal arbour. Even low rates of spontaneous vesicle release from a small fraction of GAD-utilising GABAergic neurons might summate sufficiently to provide a GABA tone at receptors on DA axons. The general function(s) of a spontaneous GABA tone are not well understood, but can serve different functions than that of action potential-dependent or synaptic events elsewhere in the brain [71,90], and could include regulation of axonal membrane resistance to modify the impact of other inputs or limit the propagation of action potentials through the axonal arbour for a sparser coding, as is suggested by the functions of tonic GABA_A_ receptor activation on limiting action potential propagation in DA axons (see Section 4).

Mesostriatal DA neurons can co-release GABA [16], which is thought to be mediated through non-canonical GABA synthesis by ALDH1a1 [83], and also through uptake of GABA from the extracellular milieu by GABA transporters suggested to be located on the plasmamembrane of DA axons [88]. GABA co-release from DA axons in striatum can evoke inhibitory currents in postsynaptic spiny projection neurons, but it is unlikely that GABA co-released from DA axons can simultaneously gate the concurrent release of DA evoked by the same discrete stimulus. Indeed, in slices pre-treated with an ALDH inhibitor (shown to attenuate GABA co-release inhibitory currents measured in postsynaptic spiny projection neurons), the tonic GABA inhibition of DA release evoked by single optogenetic stimuli is not prevented. Rather ALDH inhibition appears to enhance the tonic GABA inhibition of DA release [39]. Therefore, ALDH-dependent GABA co-released from DA axons does not appear to be responsible for the tonic inhibition of DA release, but might act indirectly to limit tonic inhibition of DA by a different, ALDH-independent, GAD-dependent source. Additionally, in a parkinsonian mouse model that exhibits enhanced tonic GABAergic inhibition of DA release in dorsal striatum (see Section 8), GABA co-release from DA axons was diminished [39], further supporting the notion the GABA co-release is unlikely to be the primary source of striatal GABA tone that acts at DA axons to inhibit DA release. However, despite the evidence against GABA co-release as a direct inhibitor of simultaneous DA release in these experiments, it remains possible that GABA co-release might modulate DA release by subsequent stimuli during more extended stimuli, as is observed for the control of DA release by presynaptic D_2_ receptors on DA axons [91]. Future studies should investigate the potential for GABA co-release to mediate feedback inhibition of DA release, and in doing so should ensure that attempts to modulate GABA co-release account for potential effects on vesicular co-storage of DA, given that GABA and DA are thought to be packaged into the same vesicle by the same vesicular monoamine transporter 2 [16,92].

Two types of striatal interneurons have recently emerged as potential sources underlying the tonic GABA receptor-dependent inhibition of striatal DA release. Striatal LTSIs are capable of autonomous firing in striatal slices [93] and a recent study has uncovered that LTSI synapses co-localize with TH-immunoreactive fibres in close proximity and that their activity attenuates DA release in dorsal striatum in a GABA_B_ receptor-dependent manner [94]. Furthermore, a sub-population of tonically active ChIs that are GAD-positive are thought to co-release GABA [17], although a recent study which tested pairs and multiple simultaneous recordings of ChIs found little evidence for GABA co-release from ChIs, at least not onto neighbouring ChIs [95]. These neuronal sources might contribute to the tonic GABA inhibition of DA release, either directly through putative synaptic inputs in close proximity to DA axons, or indirectly by contributing to the pre-existing ambient GABA tone in striatum.

There is evidence that astrocytes might release GABA in other brain regions [96,97], and in striatum astrocytes are proposed to release GABA and contribute to the ambient GABA tone through release via the Best1 channel [98] and/or via reverse transport by astrocytic GAT-3 [89]. Astrocytes are known to synthesise GABA from putrescine through a non-canonical GABA synthesis pathway involving monoamine oxidase B and ALDH1a1 synthesis enzymes [97,98,99]. However, in striatum, ALDH inhibition did not prevent GABA-receptor tonic inhibition of DA release and furthermore, GABAergic tonic inhibition of DA release is limited by and not driven by GATs located on astrocytes [39]. Consequently, astrocytes do not seem to be critical as GABA-synthesising or GABA-releasing sources of the GABA that inhibits DA release. Rather, astrocytes play a particularly important role in limiting the tonic GABA inhibition of DA release via uptake of extracellular GABA through astrocytic GATs (see Section 7).

## 7. Astrocytic GABA Transporters Set the Tone of GABAergic Inhibition of DA Release

GABA transmission across the mammalian brain is limited by uptake via plasma membrane GATs, which terminate inhibitory synaptic transmission, limit spillover to extrasynaptic sites or neighbouring synapses, and maintain GABA homeostasis to prevent excessive tonic activation of synaptic and extrasynaptic receptors [72]. Within striatum, two isoforms of GAT, namely GAT-1 (*Slc6a1*) and GAT-3 (*Slc6a11*), determine striatal ambient GABA tone. They limit tonic GABA inhibition of striatal neurons e.g., GABA_A_ receptor-mediated inhibitory conductances detected in SPNs [87,89,100,101,102], and furthermore, we recently revealed that striatal GATs also support DA release in the dorsal striatum by limiting tonic GABA inhibition of DA release [39]. We found that GAT-1 and GAT-3 individually and cooperatively limit GABA action on DA axons in dorsal striatum, thereby indirectly supporting DA release.

Within the striatum, GAT-1 is abundant on axons of GABAergic neurons [103,104,105,106], while GAT-3 is found moderately [105,106,107] but observed particularly on astrocytes [100,105,108]. Transcriptomic data have suggested that striatal astrocytes additionally express GAT-1 [108,109,110], which we recently demonstrated with GAT-1 and GAT-3 immunoreactivity on the plasmamembrane [39]. This observation challenges the long-held generalisation that GAT-1 expression is exclusively neuronal [111]. Recent work has highlighted GATs on striatal astrocytes as powerful regulators of tonic GABAergic inhibition of SPNs and striatal-dependent behaviour [100]. We therefore hypothesised that striatal astrocytes might have significant impact on DA release, and indeed observed that astrocytic GATs play a critical role in limiting the tonic inhibition of DA release in dorsal striatum through supporting GABA uptake. Metabolic inhibition of astrocytes increased tonic GABAergic inhibition DA release, therefore attenuating DA output, while also preventing the DA-attenuating effects of GAT inhibitors [39]. Our work therefore extends the emerging literature that implicates astrocytes as regulators of striatal neural activity [100,112,113,114], highlighting a previously unappreciated role for astrocytes in supporting striatal DA release. These collective findings point to GATs and astrocytes as powerful regulators of striatal and DA function.

GABAergic inhibition of DA release is observed across dorsal-ventral striatal territories [34,36,37,39], yet we made some observations that indicate a heterogeneity in GABAergic inhibition between striatal regions. We observed that GATs limit GABAergic inhibition of DA release and tonic inhibition of SPNs in dorsal striatum to a markedly greater extent than in NAc core [39]. GATs also limit GABAergic inhibition of DA release in NAc shell. Correspondingly, we found higher densities of GAT-1 and GAT-3 immunoreactivity in dorsal striatum and NAc shell than in NAc core [39]. Intriguingly, while GAT levels tally with GAT function in limiting GABA tone across striatal territories, they do not tally with the effect size of GABA-receptor antagonists on DA release in each region; higher GAT expression in dorsal striatum than NAc did not correspond to a lesser impact of GABA-receptor antagonists on DA release. Higher GAT expression levels need not necessarily predict a lower GABA tone as GATs do not simply operate continuously to lower extracellular GABA concentrations to zero, but rather function to reduce extracellular GABA concentration to a finite, non-zero value at which an equilibrium is reached. The equilibrium for GAT is not thought to be static but instead varies continuously as the driving force for GAT changes; therefore, even small changes in neuronal activity can, by alterations in Na^+^ or Cl^−^ gradients or astrocyte membrane potential, change the driving force or direction of the electrogenic GABA transport and thereby influence the extracellular GABA concentration, and consequently DA release (for review see [115]). Other mechanisms governing GABA tone might also vary between regions to sustain similar levels of GABAergic inhibition of DA release across striatum. These mechanisms might include regional differences in rate and source of GABA release, and in GABA receptor types and density. Future studies are needed to identify the mechanisms mediating divergent GABAergic regulation of DA release across striatal regions. Regardless, we find that astrocytic GATs are major regulators of ambient GABA tone in dorsal striatum, and by extension, might play an important role in facilitating dorsal-specific DA-dependent behaviours, i.e., locomotor activity and motor learning.

## 8. Clinical Implications for GABAergic Regulation of Striatal DA Release

GABAergic regulation of striatal DA release provides an axis for potential dysfunction in diseases associated with the dysregulation of DA signalling and a novel site of action for drugs to impact on DA release. In this section, we will discuss how dysfunction in GABA transporters and/or striatal GABA signalling results in dysregulated DA signalling in neurodegenerative disorders. We will further discuss how compounds that target the GABA system, such as benzodiazepines, might modify DA signalling through direct action at GABA receptors on DA axons and/or modifying striatal GABA levels upstream of GABA receptors.

Dysregulation of GATs and tonic GABA inhibition has been observed in mouse models of Parkinson’s disease. A study in external globus pallidus of DA-depleted rodents found elevated extracellular GABA and increased tonic inhibition of principal globus pallidus neurons [116]. These changes resulted from the downregulation of GAT-3 on astrocytes, mediated through a loss of DA signalling at D_2_ receptors [116]. In a highly physiological, human α-synuclein-overexpressing parkinsonian mouse model, which shows early deficits in DA release in dorsal striatum but not ventral striatum prior to overt degeneration of DA neurons in old age, an accompanying augmentation of tonic GABA inhibition of DA release occurs in early adulthood in the dorsal but not the ventral striatum [39]. This augmentation of tonic GABA inhibition was a result of downregulated levels of GAT-1 and GAT-3 at least partially located on astrocytes [39,100] in the dorsal striatum. Striatal GATs and astrocytes therefore represent novel loci for maladaptive changes in early parkinsonism that could provide novel therapeutic avenues for upregulating DA signalling in Parkinson’s disease. These observations also provide candidate explanations for why some anti-epileptics used to increase GABA function are reported to have parkinsonian-like and other motor side effects [117]; these side effects might be mediated through attenuated DA signalling in caudate-putamen.

Disturbances to striatal GATs and ambient GABA tone have also been observed in mouse models of Huntington’s disease (HD); GAT-3 is upregulated on striatal astrocytes in mouse models of HD [100,118], resulting in diminished tonic inhibition of SPNs [89,100]. The early hyperkinetic stages of HD are associated with increased DA levels in the dorsal striatum and can be treated with anti-dopaminergic therapies, yet the mechanisms underlying increased striatal DA levels in HD have not been defined [119,120]. Given the recent finding that astrocytic GATs support DA output in the dorsal striatum [39], an increase in astrocytic GAT-3 in the HD striatum would be expected to diminish tonic GABAergic inhibition and boost striatal DA output.

The regulation of striatal DA release by GATs on astrocytes therefore represents a potential locus for maladaptive plasticity in both early parkinsonism and early HD, and could provide a novel therapeutic avenue for these striatal neurodegenerative diseases. Parallel changes for astrocytic transporters are also seen for other syndromes and neurotransmitters in other brain regions. A recent study found reduced tonic inhibition in pyramidal neurons in hippocampus in a mouse model of Rett syndrome arising from upregulated GAT-3 expression and activity in neighbouring astrocytes [121], while in pre-neurodegenerative β-amyloid-based mouse models of early Alzheimer’s disease, hippocampal neurons become hyperactive due to an attenuation of glutamate uptake by astrocytes [122]. Together, these emerging strands suggest that dysregulation of astrocyte transporters might be an early feature common to a range of neurodegenerative diseases.

Striatal GABA-DA interactions might offer a novel axis for therapeutic strategies for psychomotor disorders associated with dysregulated DA signalling. One potential strategy could be to target astrocytes and GATs to modify striatal ambient GABA tone and tonic inhibition of DA release. For instance, GAT inhibitors, that have long been used clinically as antiepileptics (e.g., Tiagabine) to increase GABA function [123], could be tested for their potential to diminish DA signalling and/or GAT function in Huntington’s disease. Conversely, strategies to promote GAT function might be useful where GAT function and/or DA release is attenuated i.e., Parkinson’s disease. Although no pharmacological GAT potentiators are currently available for clinical or experimental use, a reduction of astrocyte calcium signalling in striatum (via targeted expression of a plasma membrane calcium pump) has been shown to increase astrocytic GAT-3 expression and function through a *Rab11a* signalling pathway, and to decrease tonic inhibition of SPNs [100]. These strategies might also be beneficial to other basal ganglia nuclei, like the globus pallidus, where decreased astrocytic GAT-3 expression and function and consequential enhanced tonic inhibition is also observed in mouse models of Parkinson’s disease [116].

Benzodiazepines are another class of drugs which target the GABA system and might have mechanisms of action in the striatal GABA-DA axis. Benzodiazepines are positive allosteric modulators of GABA_A_ receptors and are some of the most widely prescribed drugs due to their portfolio of anxiolytic, anticonvulsant and myorelaxant properties [124]. They do however also display considerable abuse potential [125]. At the level of the VTA, benzodiazepines disinhibit DA neuron firing through action at local GABAergic interneurons [126], which in principle would result in increased DA levels in the NAc. However, at the level of the NAc, benzodiazepines are documented to reduce basal DA levels in studies employing in vivo microdialysis [127,128,129,130,131,132,133,134,135] and to attenuate evoked DA release in studies employing ex vivo and in vivo fast-scan cyclic voltammetry [37,47,136]. Direct recordings from DA axons in striatum have demonstrated that diazepam enhances the effects of striatal GABA tone at GABA_A_ receptors on DA axons, further inhibiting DA release [47]. Altogether then, benzodiazepines appear to have a dual action that impacts on striatal DA release: while benzodiazepines act at the striatal level to blunt DA release through direct action on DA axons, their actions in the VTA appear to increase the frequency of DA release events. In support of this, voltammetry data from freely moving rats show that benzodiazepines reduce the amplitude but increase the frequency of spontaneous striatal DA events [137].

The ability of drugs such as benzodiazepines to directly inhibit DA release within striatum but disinhibit DA neuron activity in midbrain to promote the frequency of striatal DA release events highlights key complexities in understanding the outcomes of, firstly, GABA-receptors on DA output, and secondly, GABAergic potentiating compounds for clinical applications. It raises the question of whether it is the amplitude or frequency of DA release events that is important for striatal function, movement and reinforcement, and for rescue in DA disorders. For clinical applications, disinhibition of DA neuron firing might be pro-dopaminergic, but perhaps lead to the addictive properties e.g., of benzodiazepines, while actions in striatum to directly inhibit DA release would be anti-dopaminergic, leading to conflicting effects. If we are to selectively modulate GABA actions in striatum rather than midbrain, it could be important to identify whether distinct GABA_A_ receptor subtypes are expressed and could be targeted on DA axons vs. somatodendritic compartments. The large molecular diversity of GABA_A_ receptors, their heterogeneous distribution in the brain [138,139], including in striatum [140,141], and potential for modulation by subtype-preferring modulators [142], could make subtypes of axonal GABA_A_ receptors attractive as pharmacological targets for modulating DA release.

## 9. Conclusions

The extensive and highly branched axons of DA neurons allow ascending midbrain signals to be powerfully modulated at the level of the striatum. Interactions between local striatal modulators and DA axons can significantly determine how activity is converted to DA output. A revival in interest in the GABAergic modulation of DA release has identified mechanisms by which striatal GABA governs striatal DA and how these are dysregulated in neurodegeneration and addiction.

GABA plays a role in modulating striatal DA by directly suppressing DA release through GABA receptors most likely on DA axons. Furthermore, GABA receptor antagonists can enhance DA release, suggesting that a GABA tone operates within the striatum. Several potential striatal sources might contribute to a GABAergic inhibition and/or the GABA tone that impacts on striatal DA release, and future studies should elucidate the key players, cells and receptors. GATs and astrocytes stand out as important regulators of striatal and DA function by setting GABA tone. GABAergic regulation of striatal DA release in turn offers a novel axis for dysfunction in diseases associated with the dysregulation of DA signalling. Animal models of several neurodegenerative disorders such as PD and HD reveal aberrant striatal GABA tone, and compounds that normalise GABA tone should be tested for clinical efficacy.

## Figures and Tables

**Figure 1 cells-10-00709-f001:**
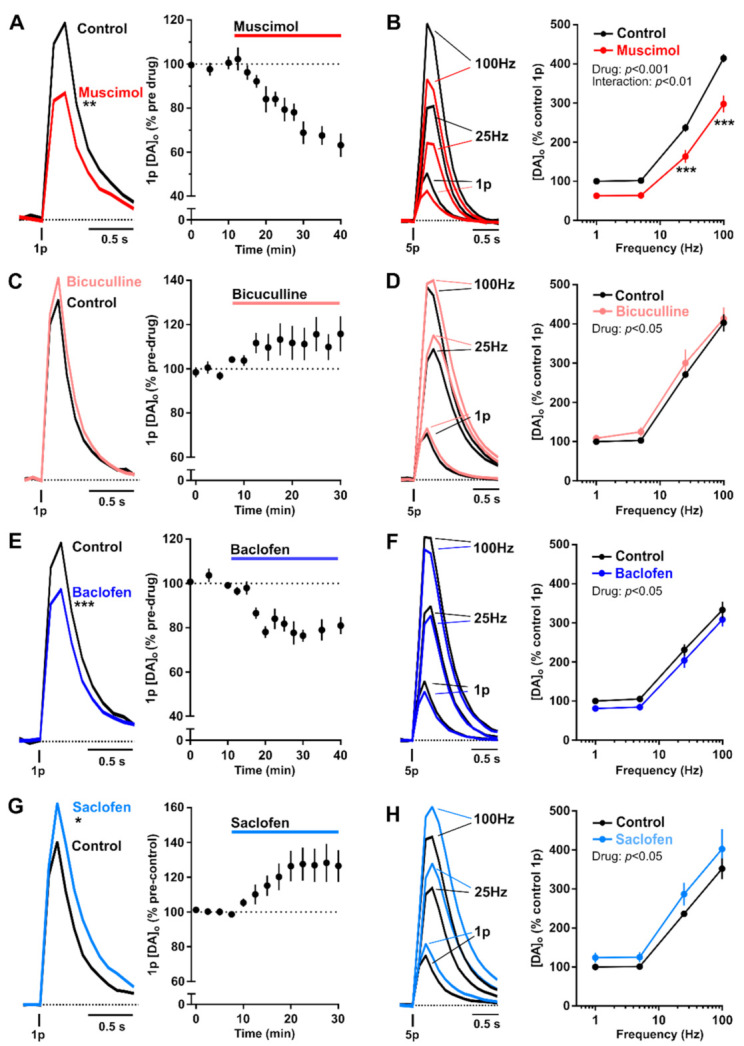
Modulation of striatal DA release in dorsal striatum by GABA_A_ and GABA_B_ receptor agonists and antagonists. (**A**, **C**, **E**, **G**)—Mean [DA]_o_ transients (left) and mean peak [DA]_o_ (± SEM) versus time (right) evoked by 1 electrical pulse (1p) in dorsal striatum in the absence (control) and presence of GABA_A_ agonist muscimol (20 μM) (**A**), GABA_A_ antagonist bicuculline (10 μM) (**C**), GABA_B_ agonist baclofen (10 μM) (**E**), or GABA_B_ antagonist saclofen (100 μM) (G). (**B**, **D**, **F**, **H**)—Mean [DA]_o_ transients (left) and mean peak [DA]_o_ (±SEM) conditions in the absence (control) and presence of GABA_A_ agonist muscimol (20 μM) (**B**), GABA_A_ antagonist bicuculline (10 μM) (**D**), GABA_B_ agonist baclofen (10 μM) (**F**), or GABA_B_ antagonist saclofen (100 μM) (**H**). All data were acquired in the presence of nAChR antagonist DHβE (1 μM). * *p* ≤ 0.05, ** *p* ≤ 0.01, *** *p* ≤ 0.001, Mann-Whitney U test (**A**,**C**,**E**,**G**), Two-way repeated measure ANOVA (**B**,**D**,**F**,**H**). Figure adapted from [36].

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
