# Peer review of "Axonal Modulation of Striatal Dopamine Release by Local γ-Aminobutyric Acid (GABA) Signalling"

_cells, 2021, doi:10.3390/cells10030709_

Round 1

Reviewer 1 Report

This manuscript by Roberts et al is an excellent of review of work relating to GABA receptor mediated influence over striatal dopamine release. The authors do an good job in writing the review which offers a largely balanced view of the topic. The writing is clear and literature search was thorough. The authors also include work from outside the striatum which helps give context. I have only minor comments:

  1. The authors mention work from Lozovaya et al 2018 that suggests cholinergic interneurons corelease GABA. It is worth mentioning that more recent work (Dorst et al 2020) tested pairs and multiple simultaneous recordings of cholinergic interneurons and found little to no evidence for GABA co-release.

  1. The abstract uses "long-standing" in the abstract and throughout which is slightly awkward. The phrase makes it sound like the information on GABA signaling was resolved long ago or is well accepted (long standing idea), but really the information is conflicting.

  1. The authors suggested that somatic GABA-A receptors generally hyperpolarize (Line 200). However, this concept has been controversial for somatodendritc GABA-A receptors as well (see Glickfeld et al NatNeuro 2009 and others).

  1. It may be useful to distinguish the function of GABA-A receptors in adults from those in developing mice which has been shown to be depolarizing in many cell types. Whether GABA-A receptors are depolarizing or hyperpolarizing has been explicitly examined in pyramidal axons (Rinetti-Vargas et al Cell Reports 2017).

  1. Another early study of inhibitory axonal GABA-A receptors in the spinal cord, perhaps the first, worth mentioning is the work of Frank and Fuortes 1957.

Reviewer 2 Report

This extremely well-written and scholarly review article provides a beautiful synthesis of a range of basic science and clinical literature.  The authors are to be congratulated on a timely and valuable contribution to the literature.  The paper should be highly cited.  A few recommendations, primarily for clarity, are summarized below.

1) It would be helpful to include data figures to illustrate the interesting point about the modest but significant change in frequency filtering of DA release with GABA-A receptor activation discussed on p. 3-4, which seems to contrast slightly with consequences of GABA-B receptor activation discussed in terms of changes in short-term plasticity on p. 6-7.

2) At first glance, the concluding sentence of Section 6 seems contradicted by evidence for a functional role of astrocytes in regulating DA release discussed in Section 7. For clarity, it would be helpful to state the distinction between these roles of astrocytes explicitly.   

Minor

Studies may “show” or “find” but do not “report” results (e.g., li. 77, 493).

li. 187. Co-release of GABA from ChIs is discussed in Section 6, not 5.

li. 241. Should add a word or two to make clear that these are general characteristics, not specific for DA axons that were discussed in the previous paragraph, e.g., “…receptors in CNS neurons can be…”.

li. 328-329. It would be appropriate to include Tritsch et al., 2014 in this list.

li. 338-340. It is a little redundant to repeat the calculated number already mentioned in the Introduction, although the point made is a good one.

References.  Titles of papers are variably capitalized or not; in ref. 18, for example, Huntington’s should have been, but is not in this non-capitalized title.

Reviewer 3 Report

In their manuscript “Axonal modulation of striatal dopamine release by local g-aminobutyric acid (GABA) signaling” BM. Roberts, EF. Lopes and SJ Cragg reviewed the literature on the modulation of striatal dopamine release by tonic GABA. This review covers an interesting topic that has, to my knowledge, not covered by any review articles before and nicely adds to a recent review by Capoche & Cheer and Liu & Kaeser on mechanisms as well as the glutamatergic and cholinergic local modulation of dopamine release. The text is nicely written, covers most relevant points and systematically guides the reader to all aspect of this topic. There are only a few point that may be considered by the authors.

Line 48: The authors may consider to provide a reference to the recent review by Capoche & Cheer to allow the interested reader to get more information about cholinergic and glutamatergic local modulation.

Line 35 “reaching an average of 2.7% of striatum” sounds strange to me. Please consider to rephrase, e.g. “reaching/spanning on average to 2.7% of the striatal volume”

Line 55: The link of “In addition” is not that clear, as in the previous sentence you did not speak about GABA release, but only about GABAergic neurons.

Line 84. Please consider to be more specific here, if memory serves these studies use only GABA(A) antagonists.

Line 115: Please consider to be more specific here, either “DA release” or “DA concentration”.

Line 121: Consider rephrasing “Altogether, these results show/demonstrate/suggest”. If “Altogether then,” is correct English, please apologize, I’m not a native speaker.

Line 124: Why the term “and so”? The release probability and the number of release sites are independent parameters of the binomial model.

Line 167: “We previously explored”. Here the reference of “we” is not clear to me.

Line 251: “However, the particular striatal GABA(A) receptor subunits presented in striatal DA axons have not yet been identified and could be different from those expressed at the soma level.” Can one really state this? Of course, there is local translation in axons. However, transcription only occurs in the nucleus, so if mRNA for GABA delta subunit would be expressed to be packed and transported to distal targets for translation, in my opinion there should be mRNAs detectable in the soma.

Line 318: “To date, …. in rats (Kubota et al., 1987)”. As this mechanism is the most obvious one to explain the effect of GABA agonists and antagonists on DA release, you may consider to provide this information earlier in the manuscript.

Line 396: “Consequently, astrocytes … that inhibits DA release”. I could not completely follow this argument, as astrocyte are a main uptake element for GABA in the brain and thus the cytosolic GABA concentration is set mainly by external GABA uptake (rather than internal synthesis). In addition, the GABA transporter acts close to electrochemical equilibrium, so the GABA tone is provided by the electrochemical properties of this transporter (see e.g. Richerson and Wu, 2003 for review). In consequence, even small changes in the neuronal activity can, by alterations in Na+, K+ gradients or glial membrane potential, change the direction of the electrogenic GABA transport and thereby influence the extracellular tonic GABA concentration (and thus the DA release). This mechanism should also be considered for your discussion in subchapter 7.

Line 442: “higher GAT expression in dorsal striatum than NAc did not correspond to a lesser impact of GABA-receptor antagonists on DA release”. See my previous comment; a higher expression of GAT must not necessarily predict a lower tonic GABA concentration.

Line 539: Because there are several GABAergic modulators described that showed some preference for putative synaptic GABAergic receptor combinations, you may speculate here about the potential use of such, more specific modulators.
